# Early changes in pulmonary function and intrarenal haemodynamics and the correlation between these sets of parameters in patients with T2DM

He Tai[1,2☯], Xiao-lin Jiang[3☯], Jin-song Kuang[4], JJ JiaJia Yu[1], Ye-tao Ju[1], Wen-cong Cao[1], Wei Chen[1], Xin-yue Cui[1], Li-de Zhang[3], Xin Fu[1], Lian-qun Jia[1]*, Yi Zhang[5]*

**1** Key Laboratory of Ministry of Education for Traditional Chinese Medicine Visera-State Theory and Application, Liaoning University of Traditional Chinese Medicine, Shenyang, China, **2** Department of Endocrinology and Metabolic, Liaoning Provincial Corps Hospital of Chinese People's Armed Police Forces, Shenyang, China, **3** Chinese and Western Medical Association College, Liaoning University of Traditional Chinese Medicine, **4** Department of Endocrinology and Metabolic, Shenyang the Fourth Hospital of People, Shenyang, China, **5** Department of Geriatrics, Shengjing Hospital of China Medical University, Shenyang, China

☯ These authors contributed equally to this work.
* jlq-8@163.com (LQZ); cmuzhangyi@163.com (YZ)

**Data Availability Statement:** All relevant data are within the Supporting Information files.

## Abstract

### Purpose

The main objectives of this study were to assess the early changes in pulmonary function and intrarenal haemodynamics and to determine the correlation between pulmonary function and intrarenal haemodynamics in patients with type 2 diabetes mellitus (T2DM).

### Methods

96 patients with T2DM (diabetes group) without diabetes kidney disease (DKD) and 33 healthy subjects (control group) were enrolled in studies intended to assess the early changes in pulmonary function and intrarenal haemodynamics associated with diabetes, as well as to determine the correlation between pulmonary function and intrarenal haemodynamics.

### Results

Pulmonary functional parameters were negatively correlated with HbA1c levels and diabetes duration ($P < 0.05$). Moreover, renal functional parameters were positively correlated with HbA1c levels and diabetes duration ($P < 0.05$). Additionally, pulmonary functional parameters were negatively correlated with renal functional parameters ($P < 0.05$). Multiple linear regression analysis of the relationship between pulmonary functional parameters and the bilateral kidney arterial resistivity index (RI) showed that all the pulmonary functional parameters were significantly correlated with the arterial RI ($P < 0.05$).

**Funding:** The study was supported by the Natural Science Foundation of China 81774022.

**Competing interests:** The authors have declared that no competing interests exist.

**Abbreviations:** 2hPBG, 2-hour postprandial blood glucose; AER, albumin excrete rate; AGEs, advanced glycated end-products; BP, blood pressure; BUN, blood urea nitrogen; CDU, color doppler ultrasound; CMECs, cardiac microvascular endothelial cells; Cr, creatinine; DBP, diastolic blood pressure; DKD, diabetes kidney disease; DLCO, diffusing capacity for carbon monoxide of lung; DLCO/VA, diffusing capacity for carbon monoxide of lung/unit volume; DN, diabetes nephropathy; DR, diabetes retinopathy; EDV, end-diastolic velocity; ESRD, end-stage renal disease; FBG, fasting plasma glucose; FEV1, forced expiratory volume in 1 second; FEV1/FVC, forced expiratory volume in 1 second/ forced vital capacity; FVC, forced vital capacity; GFR, glomerular filtration rate; HbA1c, glycosylated hemoglobinA1c; HDL-C, High-density lipoprotein cholesterol; K/DOQI, kidney Disease Outcome Quality Initiatives; LDL-C, Low-density lipoprotein cholesterol; LS, least squares; MVV, maximal voluntary ventilation; OAPP, advanced oxidation protein products; OGTT, oral glucose tolerance test; PEF, peak expiratory force; PSV, peak systolic velocity; RAS, renin-angiotensin system; RBS, random blood sugar; RI, resistivity index; SBP, systolic blood pressure; SD, standard deviation; T2DM, type 2 diabetes mellitus; TC, Total cholesterol; TG, Triglycerides; TLC, total lung capacity; UACR, urinary albumin/creatinine ratio; VC, vital capacity.

## Conclusions

Patients displayed changes in pulmonary function and intrarenal haemodynamics during the preclinical stages of DKD. Regulating glycaemia may improve intrarenal haemodynamics in the bilateral interlobular renal arteries. Moreover, during the preclinical stages of DKD, the right kidney RI is a effective predictor of early changes in pulmonary function in adult T2DM patients.

## Trial registration

ClinicalTrials.gov (NCT02798198); registered 8 June 2016.

## Introduction

The prevalence of type 2 diabetes mellitus (T2DM) is increasing worldwide, particularly in Asian countries [1]. T2DM has been identified as an independent risk factor for cardiovascular disease, as affected patients have a two-fold higher risk of developing cardiovascular disease than unaffected patients [2], and leads to the development of vascular diseases such as diabetes nephropathy (DN) and diabetes retinopathy (DR) [3], which are the leading causes of end-stage renal disease (ESRD) and acquired blindness, respectively [4]. New terminology describing kidney disease attributable to diabetes has been introduced in recent guidelines (National Kidney Foundation, 2007), which stipulate that the term 'DN' should be replaced by 'diabetes kidney disease (DKD)'.

More than 3 decades ago, researchers established that patients with T2DM had less alveolar gas exchange capacity than healthy subjects [5]. However, hyperglycaemia-induced pulmonary vascular injury is a complication of T2DM that has been overlooked by researchers attempting to devise treatments for the numerous complications associated with the disease [6].

Colour doppler ultrasound (CDU), a modality that is widely used in a variety of medical fields, evaluates blood flow velocity based on shifts in Doppler signals [7]. Patients with hyperglycaemia and uncontrolled blood pressure have significantly increased blood flow compared to patients with systemic hypertension without diabetes [8]. The Doppler resistivity index (RI) [peak systolic velocity (PSV) – end-diastolic velocity (PED)/PSV] that reflects intrarenal vascular resistance has been widely used to quantify the alterations in renal blood flow that may develop in renal disease [9].

To the best of our knowledge, no studies have investigated the correlation between pulmonary function and intrarenal haemodynamics in patients with T2DM with normal renal function. We believe that it is important to investigate the correlation between pulmonary functional parameters and intrarenal haemodynamics, as understanding this relationship may enable us to predict and prevent complications of diabetes and predict changes in pulmonary function and intrarenal haemodynamics in adults.

## Materials and methods

### Subjects

We selected 115 patients (60 males and 55 females) with T2DM without DKD, who were enrolled in a diabetes group, and 38 healthy subjects (21 males and 17 females), who were enrolled in a control group, from the diabetes outpatient clinic of the Fourth People's Hospital in Shenyang (From August 2016 to October 2016). All the participants were Han Chinese. All

the patients with T2DM were instructed to participate in a diet and exercise program developed by professional nutritionists.

## Study design

A total of 115 patients with TD2M and 38 healthy subjects, all of whom provided urine samples for analysis (AER and UACR) to exclude DKD, were recruited for the single time-point measurement phase of the study. After the above analysis, 12 patients with T2DM who were found to have DKD were excluded from the study. Additionally, 3 healthy subjects suffering from colds were also excluded from the study. Thus, 103 patients with T2DM and 35 healthy subjects were eligible to participate in subsequent tests. The subjects fasted for at least 8 hours, after which 5 ml of venous blood was collected from each participant. The subjects also underwent pulmonary function tests, intraocular pressure measurements, and retrobulbar haemodynamics (RI) assessments, the latter of which were performed by CDU. We could not locate the interlobular renal arteries in 7 patients with T2DM and 2 healthy subjects (5.71%) with urolithiasis, who were subsequently excluded from the study. Thus, 96 patients with T2DM (52 males, 44 females) and 33 healthy subjects (18 males, 15 females) ultimately completed this phase of the study.

The two groups did not differ significantly ($P > 0.05$) with respect to the sex ratio or age (range, 35 to 66 years). Data pertaining to diabetes durations (range, 4 to 12 years) and HbA1c values (range, 7.0% (53 mmol/mol) to 10% (86 mmol/mol)), FBG (range, 6.8 to 9.2 mg/dl), 2 hPBG (range, 9.2 to 13.6 mg/dl), the indicated pulmonary functional parameters (VC% (range, 73 to 90%), FVC% (range, 70 to 84%), FEV1% (range, 70 to 85%), PEF% (range, 47 to 73%), MVV% (range, 75 to 91%), TLC% (range, 81 to 99%), the FEV1/FVC% ratio (range, 70 to 82%), DLCO% (range, 74 to 90%), and the DLCO/VA% ratio (range, 74 to 92%)), the indicated intrarenal haemodynamics parameter (bilateral kidney resistivity index (RI)) (range, 0.6 to 0.7), and the indicated blood lipid (TC (range, 178 to 236 mg/dl), HDL-C (range, 31 to 48 mg/dl), LDL-C (range, 102 to 142 mg/dl), and TG(range, 138 to 173 mg/dl)) and renal functional parameters (BUN (range, 3.6 to 7.0 mmol/L), Cr (range, 31 to 62 μmol/L), and GFR (range, 110 to 150 ml/minute)), as well as data pertaining to the AER (range, 12 to 18 μg/min), the UACR (range, 13 to 18 mg/g) and BP (SBP (range, 112 to 138 mmHg), DBP (range, 75 to 100 mmHg), were recorded (Table 1). T2DM and DKD was diagnosed in accordance with the guidelines of the American Diabetes Association [10].

## Study assessments and endpoints

Blood specimen collection and laboratory testing: Venous blood samples were collected between 6 and 8 AM following a fasting period of at least 8 hours and used for measurements of FPG levels, HbA1c levels, blood lipid levels, and renal functional parameters. Plasma glucose levels were determined by the glucose oxidase method. Venous blood samples were collected to measure 2-hPG levels. They were measured according to the instructions of the corresponding research kits.

Urine sample collection and laboratory tests: Urine output was quantified via a single 24-h urine collection. Urinary albumin concentrations were measured using a double-antibody radio immunoassay with a sensitivity of 0.5 mg/l, an intra-assay coefficient of variation of 4.5%, and an interassay coefficient of variation of 5.3% within the range of 10–50 mg/l [11].

Blood pressure (BP) measurements: systolic blood pressure (SBP) and diastolic blood pressure (DBP) were measured using an electronic sphygmomanometer.

Pulmonary function measurements: The indicated pulmonary functional parameters (VC%, FVC%, FEV1%, PEF%, MVV%, TLC%, the FEV1/FVC% ratio, DLCO%, and the DLCO/

**Table 1. Baseline demographic and clinical characteristics.**

| Characteristics | | Control group | Diabetes group | $t/\chi^2$ value | P value* |
|---|---|---|---|---|---|
| NO. (*n*) | | 33 | 96 | - | - |
| Sex, *n* (%) | Male | 18 (54.5) | 52 (54.2) | 0.001 | 0.970 |
| | Female | 15 (45.5) | 44 (45.8) | | |
| Age (years) | | 51.73±7.99 | 52.59±9.06 | 0.423 | 0.674 |
| Diabetes duration (years) | | - | 7.92±2.17 | - | - |
| FBG (mmol/l) | | 4.97±0.55 | 7.88±0.63 | 20.334 | < 0.0001* |
| 2 hPBG (mmol/l) | | 7.20±0.31 | 11.53±0.93 | 25.499 | < 0.0001* |
| HbA1c | % | 5.19±0.57 | 8.04±0.85 | 16.303 | < 0.0001* |
| | mmol/mol | 46.91±16.69 | 65.77±8.52 | 6.049 | < 0.0001* |
| Baseline HbA1c, *n* (%) | ≤ 7%/ (53 mmol/mol) | 33 (100.0) | 6 (16.2) | 49.63 | < 0.0001* |
| | > 7%/ (53 mmol/mol) | 0 (0.0) | 31 (83.8) | | |
| TC (mg/dl) | | 188.22±14.85 | 202.97±14.70 | 4.113 | < 0.0001* |
| HDL-C (mg/dl) | | 45.64±4.68 | 37.89±4.34 | -7.187 | < 0.0001* |
| LDL-C (mg/dl) | | 112.06±9.22 | 122.19±9.87 | 4.422 | < 0.0001* |
| TG (mg/dl) | | 142.21±8.85 | 153.03±8.37 | 5.251 | < 0.0001* |
| Cholesterol-lowering drugs *n* (%) | use | 6 (18.2) | 15 (40.5) | 4.152 | 0.042 |
| | no use | 27 (81.8) | 22 (59.5) | | |
| SBP (mmHg) | | 124.70±6.72 | 127.54±5.96 | 1.876 | 0.065 |
| DBP (mmHg) | | 88.97±5.07 | 89.97±5.71 | 0.774 | 0.442 |
| Cr (μmol/L) | | 64.64±7.67 | 77.97±8.14 | 7.032 | < 0.0001* |
| BUN (mmol/L) | | 4.79±0.61 | 6.13±0.72 | 8.361 | < 0.0001* |
| GFR (ml/minute) | | 119.70±6.96 | 134.19±5.46 | 9.746 | < 0.0001* |
| AER (μg/min) | | 7.97±1.69 | 16.22±1.27 | 23.25 | < 0.0001* |
| UACR (mg/g) | | 7.82±1.33 | 15.68±1.18 | 26.16 | < 0.0001* |
| Mean right kidney RI | | 0.61±0.02 | 0.65±0.02 | 8.089 | < 0.0001* |
| Mean left kidney RI | | 0.61±0.02 | 0.65±0.02 | 7.765 | < 0.0001* |
| Mean Doppler RI | | 0.61±0.02 | 0.65±0.02 | 11.240 | < 0.0001* |
| VC Litre (% of predicted) | | 87.30±4.28 | 81.81±4.20 | -5.414 | < 0.0001* |
| FVC Litre (% of predicted) | | 79.12±3.80 | 76.08±3.14 | -3.665 | < 0.0001* |
| FEV1 Litre (% of predicted) | | 77.88±2.42 | 75.35±2.90 | -3.932 | < 0.0001* |
| PEF L/S (% of predicted) | | 58.21±2.88 | 51.62±2.63 | -10.009 | < 0.0001* |
| MVV Litre (% of predicted) | | 89.21±2.13 | 85.73±2.28 | -6.574 | < 0.0001* |
| TLC Litre (% of predicted) | | 95.39±2.78 | 91.35±3.75 | -5.069 | < 0.0001* |
| FEV1/FVC (% of predicted) | | 77.85±2.95 | 74.19±3.03 | -5.111 | < 0.0001* |
| DLCO (mL/min/mmHg) (% of predicted) | | 87.15±2.73 | 84.86±2.94 | -3.362 | 0.001* |
| DLCO/VA(mL/min/mmHg) (% of predicted) | | 87.89±2.83 | 90.76±2.45 | -4.506 | < 0.0001* |

Note: FBG, fasting plasma glucose; 2hPBG, 2-hour postprandial blood glucose; HbA1c, glycosylated hemoglobinA1c; TC, Total cholesterol; HDL-C, High-density lipoprotein cholesterol; LDL-C, Low-density lipoprotein cholesterol; TG, Triglycerides; SBP, systolic blood pressure; DBP, diastolic blood pressure; BUN, blood urea nitrogen; Cr, creatinine; UACR, urinary albumin/creatinine ratio; AER, albumin excrete rate; GFR, glomerular filtration rate; VC, vital capacity; FVC, forced vital capacity; FEV1, forced expiratory volume in 1 second; PEF, peak expiratory force; MVV, maximal voluntary ventilation; TLC, total lung capacity; FEV1/FVC, forced expiratory volume in 1 second/ forced vital capacity; DLCO, diffusing capacity for carbon monoxide of lung; DLCO/VA, diffusing capacity for carbon monoxide of lung/unit volume

*P< 0.05, Statistically significant

VA% ratio) were measured using a spirometer. We used the measured-to-expected value ratios and the percentages of the predicted value to eliminate the influence of age, height, and weight on the results.

Intrarenal haemodynamic parameter measurements: We measured the indicated intrarenal haemodynamics parameters (PSV, EDV, and RI) in the bilateral interlobular renal arteries of subjects. The pulsed Doppler sampling gate was located in the interlobar arteries, and the angle of insonation. The PSV and EDV were documented in centimetres per second, and the RI was calculated [12].

## Statistical analysis

Measurement data were expressed as the mean ±standard deviation (SD), and numerical data were expressed as percentages. Statistical analysis was conducted using the SPSS statistical package (Version 17.0, SPSS Inc. Chicago, IL, USA). The linear correlations between pulmonary function and HbA1c levels and diabetes duration, the linear correlations between intrarenal haemodynamics and HbA1c and diabetes duration and the linear correlations between pulmonary function and intrarenal haemodynamics were evaluated using Pearson's correlation coefficient. For multiple linear regression analysis, the pulmonary functional parameters were the dependent variables, and the bilateral kidney RIs were the independent variables. $P <$ 0.05 was considered statistically significant.

## Results

### Participant baseline characteristics

At last, 96 subjects (52 males and 44 females) were enrolled in the diabetes group, and 33 healthy people (18 males, 15 females) were enrolled in the control group.

Pulmonary functional parameters, FPG, 2 hPG, HbA1c, TC, LDL-C, TG, Cr, and BUN levels; the GFR, AER, and UACR; and the bilateral RIs were significantly greater in the diabetes group than in the control group ($P <$ 0.05). HDL-C levels were lower in the diabetes group than in the control group ($P <$ 0.05); however, SDP and DBP were not significantly different between the two groups ($P >$ 0.05) (Table 1).

### Correlation between pulmonary functional parameters and HbA1c and diabetes duration

Pulmonary functional parameters were significantly negatively correlated with HbA1c and diabetes duration ($P <$ 0.05) (Table 2); however, in the control group, the correlation between pulmonary functional parameters and HbA1c was not statistically significant ($P >$ 0.05).

### Correlation between renal functional parameters and HbA1c and diabetes duration

Renal functional parameters were significantly positively correlated with HbA1c and diabetes duration ($P <$ 0.05) (Table 3); however, in the control group, the correlation between renal functional parameters and HbA1c was not statistically significant ($P >$ 0.05).

### Correlation between pulmonary functional parameters and renal functional parameters

Pulmonary functional parameters were significantly negatively correlated with renal functional parameters ($P <$ 0.05) (Table 4); however, in the control group, the correlation between

**Table 2. Correlation of pulmonary function parameters to HbA1c and diabetes duration in diabetes group.**

| Pulmonary function parameters | HbA1c | Diabetes duration |
|---|---|---|
| VC | $r$ = -0.161 | $r$ = -0.690* |
| P value | 0.342 | < 0.0001 |
| FVC | $r$ = 0.250 | $r$ = -0.213 |
| P value | 0.135 | 0.205 |
| FEV1 | $r$ = -0.017 | $r$ = -0.024 |
| P value | 0.918 | 0.886 |
| PEF | $r$ = -0.132 | $r$ = -0.177 |
| P value | 0.435 | 0.294 |
| MVV | $r$ = -0.168 | $r$ = -0.511* |
| P value | 0.321 | 0.001 |
| TLC | $r$ = -0.212 | $r$ = -0.373* |
| P value | 0.208 | 0.023 |
| FEV1/FVC | $r$ = -0.129 | $r$ = -0.394* |
| P value | 0.447 | 0.016 |
| DLCO | $r$ = -0.185 | $r$ = -0.303 |
| P value | 0.272 | 0.068 |
| DLCO/VA | $r$ = -0.306 | $r$ = -0.329* |
| P value | 0.066 | 0.047 |

Note:

*$P$< 0.05, Statistically significant

HbA1c, glycosylated hemoglobinA1c; VC, vital capacity; FVC, forced vital capacity; FEV1, forced expiratory volume in 1 second; PEF, peak expiratory force; MVV, maximal voluntary ventilation; TLC, total lung capacity; FEV1/FVC, forced expiratory volume in 1 second/ forced vital capacity; DLCO, diffusing capacity for carbon monoxide of lung; DLCO/VA, diffusing capacity for carbon monoxide of lung/unit volume

pulmonary functional parameters and renal functional parameters was not statistically significant ($P$> 0.05). Multiple linear regression analysis showed that all nine pulmonary functional parameters assessed herein were significantly correlated with the bilateral kidney RI ($P$< 0.05). (Table 5).

## Discussion

Renal Doppler RIs are widely used to evaluate blood flow in renal parenchymal diseases. RIs, which reflect intrarenal vascular resistance, a measure of vascular compliance [9], are also widely used to quantify changes in renal blood flow that may be attributable to renal diseases. Interestingly, previous studies have shown that renal Doppler sonography is an effective non-invasive and inexpensive means of screening for renovascular hypertension correctable via treatment with captopril [13]; therefore, we evaluated intrarenal haemodynamics by examining the RI using Doppler sonography.

HbA1c is an indicator of diabetes control; thus, the higher the HbA1c level, the poorer the diabetes control and the higher the circulating glucose concentration. Persistent elevations in circulating glucose levels (as measured by HbA1c) lasting 3 months or longer can lead to increases in nonenzymatic tissue protein glycosylation [14]. The findings of this study regarding HbA1c levels were consistent with those of other studies despite the fact that the current study enrolled patients with HbA1c levels as low as 7.0% (53 mmol/mol) [14]. Prabhu M *et al* [15] showed that only 23 (11.5%) patients enrolled in their study were aware of the importance

**Table 3. Correlation of the renal parameters to HbA1c and diabetes duration in diabetes group.**

| Renal parameters | HbA1c | Diabetes duration |
|---|---|---|
| Cr | $r = 0.727^*$ | $r = 0.046$ |
| P value | < 0.0001 | 0.789 |
| BUN | $r = 0.050$ | $r = 0.151$ |
| P value | 0.769 | 0.373 |
| GFR | $r = 0.635^*$ | $r = 0.291$ |
| P value | 0.000 | 0.081 |
| AER | $r = 0.017$ | $r = 0.354^*$ |
| P value | 0.922 | 0.032 |
| UACR | $r = 0.272$ | $r = 0.348^*$ |
| P value | 0.103 | 0.035 |
| Right kidney RI | $r = 0.411^*$ | $r = 0.847^*$ |
| P value | 0.012 | 0.000 |
| Left kidney RI | $r = 0.565^*$ | $r = 0.738^*$ |
| P value | 0.000 | 0.000 |

Note:

$^*P< 0.05$, Statistically significant

BUN, blood urea nitrogen; Cr, creatinine; UACR, urinary albumin/creatinine ratio; AER, albumin excrete rate; GFR, glomerular filtration rate; HbA1c, glycosylated hemoglobinA1c; RI, resistivity index

of HbA1c levels. Moreover, the authors of that study noted that the proportion of patients who achieved the target HbA1c level of <7% (53 mmol/mol) was low in patients with DR. In our study, the proportion of patients (12/96; 12.50%) who reached this HbA1c standard [<7.0% (53 mmol/mol)] was also low, as most of the patients enrolled herein could not control their glycaemia adequately; thus, the mean FPG (7.88 mmol/l) and 2-hPG (11.53 mmol/l) levels in the diabetes group were higher than the corresponding target levels (7 mmol/l and 10 mmol/l respectively).

The mechanisms underlying the occurrence of lung damage in diabetes are not fully understood; however, glycaemic control appears to play a key role in the association between reductions in lung function and diabetes. In addition, nonenzymatic glycosylation of proteins in the lungs decreases lung compliance [16] and thus diminishes the large microvascular reserve of the alveolar-capillary system and increases its susceptibility oxidative damage. Thus, hyperglycaemia damages the lung [17]. Clinically, loss of microvascular reserve in the lung may be associated with an increased risk of hypoxia in acute or chronic pathological lung conditions [17].

Lung CO transfer capacity is significantly affected by the integrity of the lung capillary endothelium, a finding that supports that idea that clinicians should devote more attention to pulmonary vascular changes. The reports on lung function testing in patients with diabetes that have been published during the last 15 years have focused predominantly on pulmonary microangiopathy; however, relatively few studies have focused on pulmonary mechanical function. The lung functional parameters that are related specifically to pulmonary microangiopathy include pulmonary capillary blood volume and CO transfer capacity [18]. Niranjan V found that TLC, FVC, FEV1, and VC values were significantly lower in patients with type 1 diabetes than in healthy subjects [19]. However, we selected patients with T2DM whose diabetes durations ranged from 4 to 12 years (<15 years) and assessed a smaller population than Niranjan V [20]. The results pertaining to the correlations between HbA1c and pulmonary function that were noted in previous studies were inconsistent. Study noted weak associations

**Table 4. Correlation between pulmonary function parameters and renal parameters in diabetes group.**

| Pulmonary function | Renal parameters | | | | | | |
|---|---|---|---|---|---|---|---|
| | Cr | BUN | GFR | AER | UACR | Right RI | Left RI |
| VC | r = 0.125 | r = 0.090 | r = -0.336* | r = - .497* | r = - 0.372* | r = - 0.718* | r = - 0.535* |
| P value | 0.462 | 0.597 | 0.042 | 0.002 | 0.023 | 0.000 | 0.001 |
| FVC | r = 0.013 | r = - 0.116 | r = - 0.037 | r = - 0.046 | r = 0.105 | r = - 0.328* | r = - 0.194 |
| P value | 0.938 | 0.494 | 0.830 | 0.786 | 0.537 | 0.048 | 0.250 |
| FEV1 | r = 0.180 | r = - 0.071 | r = - 0.061 | r = - 0.021 | r = 0.010 | r = - 0.174 | r = - 0.114 |
| P value | 0.286 | 0.676 | 0.722 | 0.901 | 0.954 | 0.302 | 0.500 |
| PEF | r = - 0.172 | r = - 0.155 | r = - 0.142 | r = 0.058 | r = - 0.094 | r = - 0.050 | r = - 0.036 |
| P value | 0.308 | 0.358 | 0.402 | 0.731 | 0.578 | 0.769 | 0.831 |
| MVV | r = .100 | r = - .005 | r = - .447* | r = - .037 | r = - .240 | r = - .446* | r = - 0.343* |
| P value | 0.556 | 0.978 | 0.006 | 0.829 | 0.153 | 0.006 | 0.038 |
| TLC | r = 0.026 | r = 0.079 | r = - 0.567* | r = - 0.063 | r = - 0.124 | r = - 0.390* | r = - 0.263 |
| P value | 0.881 | 0.641 | 0.000 | 0.711 | 0.464 | 0.017 | 0.115 |
| FEV1/FVC | r = - 0.244 | r = - 0.043 | r = - 0.255 | r = - 0.047 | r = 0.049 | r = - 0.453* | r = - 0.322 |
| P value | 0.146 | 0.802 | 0.128 | 0.782 | 0.774 | 0.005 | 0.052 |
| DLCO | r = 0.020 | r = - 0.103 | r = - 0.217 | r = - 0.022 | r = - 0.053 | r = 0.327* | r = - 0.267 |
| P value | 0.908 | 0.543 | 0.197 | 0.899 | 0.755 | 0.048 | 0.111 |
| DLCO/VA | r = 0.098 | r = - 0.131 | r = - 0.009 | r = - 0.179 | r = - 0.027 | r = - 0.417* | r = - 0.368* |
| P value | 0.565 | 0.439 | 0.956 | 0.290 | 0.872 | 0.010 | 0.025 |

Note

*$P < 0.05$, Statistically significant

BUN, blood urea nitrogen; Cr, creatinine; UACR, urinary albumin/creatinine ratio; AER, albumin excrete rate; GFR, glomerular filtration rate; VC, vital capacity; FVC, forced vital capacity; FEV1, forced expiratory volume in 1 second; PEF, peak expiratory force; MVV, maximal voluntary ventilation; TLC, total lung capacity; FEV1/FVC, forced expiratory volume in 1 second/ forced vital capacity; DLCO, diffusing capacity for carbon monoxide of lung; DLCO/VA, diffusing capacity for carbon monoxide of lung/unit volume; RI, resistivity index

**Table 5. Multiple regression analysis between pulmonary function and bilateral kidney RI in diabetes group.**

| Dependent variable | Coefficient | | | | | P value* |
|---|---|---|---|---|---|---|
| | R | R² | F | Right RI | Left RI | |
| VC | 0.728 | 0.530 | 19.197 | - 0.898 | 0.218 | 0.000* |
| FVC | 0.355 | 0.126 | 2.451 | - 0.529 | 0.244 | 0.101 |
| FEV1 | 0.182 | 0.033 | 0.583 | - 0.252 | 0.094 | 0.563 |
| PEF | 0.051 | 0.003 | 0.044 | -0.063 | 0.016 | 0.957 |
| MVV | 0.448 | 0.201 | 4.270 | - 0.513 | 0.081 | 0.022* |
| TLC | 0.404 | 0.164 | 3.325 | - 0.546 | 0.188 | 0.048* |
| FEV1/FVC | 0.463 | 0.214 | 4.636 | - 0.591 | 0.167 | 0.017* |
| DLCO | 0.327 | 0.107 | 2.036 | - 0.337 | 0.012 | 0.146 |
| DLCO/VA | 0.419 | 0.175 | 3.618 | - 0.356 | 0.074 | 0.038* |

Note:

*$P < 0.05$, Statistically significant

RI, resistivity index; BUN, blood urea nitrogen; Cr, creatinine; UACR, urinary albumin/creatinine ratio; AER, albumin excrete rate; GFR, glomerular filtration rate; VC, vital capacity; FVC, forced vital capacity; FEV1, forced expiratory volume in 1 second; PEF, peak expiratory force; MVV, maximal voluntary ventilation; TLC, total lung capacity; FEV1/FVC, forced expiratory volume in 1 second/ forced vital capacity; DLCO, diffusing capacity for carbon monoxide of lung; DLCO/VA, diffusing capacity for carbon monoxide of lung/unit volume

between HbA1c and spirometric parameters and strong correlations between diabetes duration and pulmonary function [21]. Another cross-sectional population study noted that plasma glucose levels were negatively correlated with FVC and/or FEV [22]. HbA1c is an indicator of diabetes control, and our results revealed that pulmonary functional parameters were negatively correlated with HbA1c levels ($P < 0.05$); however, in healthy people, the correlations between the above parameters were not statistically significant ($P > 0.05$). The correlations between pulmonary functional parameters and diabetes duration were statistically significant ($P < 0.05$) (Table 2), results similar to those of previous studies [6, 21–22]. Notably, good glycaemic control and regular diabetes treatment have positive effects on lung function in patients with T2DM.

The precise mechanisms underlying DKD development are unknown; however, several theories exist regarding the specific processes that affect haemodynamics in DKD. Renin-angiotensin system (RAS) activation reportedly induces intrarenal haemodynamic abnormalities in diabetes, the intrarenal RAS may be activated in diabetes and subsequently facilitate increases in the RI, and blocking RAS activation with captopril may reduce intrarenal vascular resistance in diabetes [23, 24].

Elevated RIs have been reported to be associated with vascular-interstitial diseases, including DKD (but not glomerulopathies). Increased RIs may be reflective of decreased tissue and vascular compliance, as well as increased vascular resistance [9]. However, the early stages of DN are associated with an increased GFR and variable increases in renal plasma flow and the filtration fraction in both clinical and experimental settings. These changes may also be reflected by increased RIs. Our results showed that the mean GFR in the diabetes group (134.19±5.46 ml/min) was significantly greater than that in the control group (119.70±6.96 ml/min) ($P < 0.05$), results similar to those of the study by Pelliccia P and Matsumoto N [24, 25]. RIs are measured by duplex Doppler sonography, a noninvasive and inexpensive tool that is useful for demonstrating the haemodynamic abnormalities present in patients with DKD [26]. Biopsy studies involving children have shown that basement membrane thickening and mesangial expansion in the kidney develop prior to the onset of microalbuminuria [27]. To our knowledge, our study is the first to assess the early changes in intrarenal haemodynamics associated with diabetes in adults with T2DM without any evidence of renal dysfunction, and our data indicate that the RI can be used to predict changes in renal function in the preclinical stage of DKD, results similar to those of the study by Pelliccia P, which involved children [24].

However, there is no general agreement with respect to the significance and predictive value of the renal RI in patients with DKD. Researchers have performed several studies regarding the application of Doppler sonography for the evaluation of intrarenal haemodynamic abnormalities in adults with DKD [9]; however, studies regarding the preclinical stage of DKD (in which renal function is normal) in adults are still lacking. In our study, we aimed to explore whether Doppler sonography could be used to detect alterations in the renal RI in adults with diabetes who had normal renal function, according to their laboratory test results.

We observed that adults with T2DM had significantly higher RI values than healthy controls ($P < 0.05$) (Table 1); however, all the patients (in both the diabetes and control groups) had at least one RI value less than or equal to 0.70, the generally accepted cut-off value separating healthy adults and children older than 6 years (rather than younger children) from unhealthy adults and children [28]. The RI is positively correlated with HbA1c and diabetes duration (both $P < 0.05$) (Table 3); however, in the control group, the correlation between renal functional parameters and HbA1c was not statistically significant ($P > 0.05$). In the present study, all the RI values were >0.5. Additionally, we observed that BUN and Cr and the GFR, AER, and UACR were positively correlated with HbA1c and diabetes duration; however, not all of these correlations were statistically significant. Thus, we concluded that the predictive

value of the RI was greater than that of the other parameters listed above. As the sample size of the study was not large enough for the results pertaining to the other parameters to achieve sufficient statistical power, we able to analyse only the predictive value of the RI and GFR.

In our study, not all the correlations between pulmonary functional parameters and renal functional parameters were statistically significant. All the pulmonary functional parameters were significantly negatively correlated with the bilateral kidney RI ($P<$ 0.05) (Table 4); however, in the control group, the correlations between pulmonary functional parameters and the RI were not statistically significant ($P>$ 0.05). According to our results, the bilateral kidney RI may be used to evaluate the interactions between renal and pulmonary function during the preclinical stage of DKD. Moreover, multiple linear regression analysis showed that all the correlations between the pulmonary functional indices assessed herein and the bilateral kidney RI were statistically significant ($P<$ 0.05). (Table 5). Therefore, the results of our multiple linear regression analysis indicated that the right RI could be used as a strong predictor of pulmonary function in adults with T2DM with normal renal function. Why the right RI is a more powerful predictor of pulmonary function than the left RI. We surmised that the right RI was more accurate than in the left in our study because our Doppler sonographer was standing on the right side of each patient.

## Conclusions

The results of this study indicate that in addition to renal functional parameters, the combination of the right kidney RI, GFR, and HbA1c may also be good predictors of changes in pulmonary function in patients with diabetes, as well as a more sensitive indicator of changes in pulmonary function in such patients during the pre-clinical stages of DKD than the UACR or AER. The predictive value of the combination is higher than that of either parameter alone.

However, our study had several limitations that should be addressed in future studies. First, we failed to observe the changes in alveolar tissue morphology associated with diabetes and did not identify the specific protein responsible for inducing the changes. Because not all the patients underwent a lung biopsy, we had to adopt an animal model to study alveolar tissue samples. Second, we did not assess the long-term changes in pulmonary function associated with diabetes, and we studied the correlations between pulmonary and renal functional parameters only during the preclinical stage of DKD. It is worth evaluating whether they can be used to predict pulmonary function and renal function during the preclinical stage of DKD. Moreover, we studied T2DM without DKD but did not determine the correlations between the above parameters in different phases of DKD. Additionally, arterial elasticity (which influences haemodynamic parameters) decreases with increasing age; however, we did not consider age as a variable in our study. Therefore, we recommend that clinicians monitor patients with T2DM for signs of lung damage in addition to monitoring them for signs of DKD and DR.

## Supporting information

**S1 File. All data underlying the findings are described in this file.**
(XLS)

## Acknowledgments

The authors would like to thank all of the patients and their families, the team of investigators, research nurses, and operations staff involved in this study. Editorial support (in the form of writing assistance, including development of the initial draft based on author input,

assembling tables and figures, collating authors comments, grammatical editing and referencing) was provided by He Tai.

## Author Contributions

**Data curation:** Jin-song Kuang, Ye-tao Ju, Li-de Zhang, Yi Zhang.

**Formal analysis:** Yi Zhang.

**Investigation:** Wen-cong Cao, Wei Chen, Li-de Zhang, Xin Fu, Yi Zhang.

**Methodology:** Ye-tao Ju, Xin Fu, Yi Zhang.

**Resources:** Xin Fu.

**Software:** Yi Zhang.

**Supervision:** Ye-tao Ju, Xin-yue Cui, Li-de Zhang.

**Writing – original draft:** He Tai, Xiao-lin Jiang, JJ JiaJia Yu, Yi Zhang.

**Writing – review & editing:** Lian-qun Jia, Yi Zhang.

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
