## [Decision Letter · Decision Letter 0]

23 Sep 2019

PONE-D-19-16612

Early Changes in Pulmonary Function and Intrarenal Haemodynamics and the Correlation between These Sets of Parameters in Patients with T2DM

PLOS ONE

Dear Dr jia,

Thank you for submitting your manuscript to PLOS ONE. After careful consideration, we feel that it has merit but does not fully meet PLOS ONE’s publication criteria as it currently stands. Therefore, we invite you to submit a revised version of the manuscript that addresses the points raised during the review process.

We would appreciate receiving your revised manuscript by Nov 07 2019 11:59PM. To enhance the reproducibility of your results, we recommend that if applicable you deposit your laboratory protocols in protocols.io, where a protocol can be assigned its own identifier (DOI) such that it can be cited independently in the future. For instructions see: http://journals.plos.org/plosone/s/submission-guidelines#loc-laboratory-protocols

We look forward to receiving your revised manuscript.

Kind regards,

Alon Harris

Academic Editor

PLOS ONE

Journal Requirements:

4. We note that you have reported significance probabilities of 0 in places. Since p=0 is not strictly possible, please correct this to a more appropriate limit, eg 'p<0.0001

5. Please include your tables as part of your main manuscript and remove the individual files. Please note that supplementary tables (should remain/ be uploaded) as separate "supporting information" files

Additional Editor Comments (if provided):

Reviewers' comments:

Reviewer's Responses to Questions

**Comments to the Author**

1. Is the manuscript technically sound, and do the data support the conclusions?

Reviewer #1: Yes

Reviewer #2: Partly

Reviewer #3: Yes

2. Has the statistical analysis been performed appropriately and rigorously? 

Reviewer #1: Yes

Reviewer #2: Yes

Reviewer #3: Yes

3. Have the authors made all data underlying the findings in their manuscript fully available?

Reviewer #1: Yes

Reviewer #2: No

Reviewer #3: Yes

4. Is the manuscript presented in an intelligible fashion and written in standard English?

Reviewer #1: Yes

Reviewer #2: Yes

Reviewer #3: Yes

5. Review Comments to the Author

Reviewer #1: This is very well written article on a subject which of great importance. Diabetes and its associated complications can affect the quality of life of patients affected by this disease. Evaluating the pulmonary functions and the RI of the blood vessels is a novel way to assess the prognosis of Diabetes.

Few grammatical errors need to be corrected.

Reviewer #2: Early Changes in Pulmonary Function and Intrarenal Haemodynamics and the Correlation between These Sets of Parameters in Patients with T2DM

This manuscript describes pulmonary function testing and renal function testing including the kidney resistivity index in patients with type 2 diabetes and controls. The study also compares the relationship between pulmonary and renal function in diabetic patients. The grammar, introduction, and methods are clear. The results and discussion may need more clarification.

Grammar:

There were few grammatical errors requiring revision. Examples include:

Author’s page: “Corresponding autuo” should read Corresponding author

Discussion, 4th paragraph: “that idea” should read the idea

Discussion, 4th paragraph: “Study noted weak associations…” should read One study

Discussion, 8th paragraph: “sufficient statistical power, we able…” should read we were able

Introduction:

The introduction explains the topic well, provides justification for the study, and cites existing literature. One point of clarification would be helpful:

Introduction, paragraph 4: “no studies have investigated the correlation between pulmonary function and intrarenal haemodynamics in patients with T2DM with normal renal function.” Are you stating that in your study, the diabetics had normal renal function? The discussion section references table 3 showing that the diabetics had increased GFR relative to controls.

Methods:

The methods section is very good. It includes sufficient information on the IRB/consent, protocol, inclusion/exclusion of patients, instruments used, and calculations. However, further clarification would be helpful in the following areas:

Study design, paragraph 1: “retrobulbar haemodynamics” I believe this is in error and should read intrarenal hemodynamics.

Results:

The results section is well organized and easy to read. However, further clarification would be helpful in the following areas:

Abstract and Results section: The results for renal function parameters are confusing and may be worded better:

Upon initial reads, stating that the renal function parameters are positively correlated with HbA1c and diabetes duration sounds like renal function gets better with diabetes, which is not the case, nor the point of your manuscript. You very nicely show that RI increases with both increased HbA1c and diabetes duration, which I believe is more clear.

In addition, when stating that pulmonary function and renal function are negatively correlated, it once again falsely sounds like the lungs do worse in diabetes while the kidneys get better. It would be clearer to state that the pulmonary function is negatively correlated with kidney RI, GFR, and HbA1C. I believe you state elsewhere in the paper that many of the other renal parameters were not statistically significant anyhow.

Discussion:

The discussion section reviews and analyzes the current study. It cites relevant literature. Limitations of the study are discussed as well as areas for future research. It explains findings well regarding pulmonary function. However, there are areas where further clarification of the renal findings would be helpful:

Discussion, paragraph 6: “our study is the first to assess the early changes in intrarenal haemodynamics associated with diabetes in adults with T2DM without any evidence of renal dysfunction…” You stated in the same paragraph that the GFR was higher in the diabetics. So, there is evidence of renal dysfunction? Could the increased GFR in early diabetes be a predictor of renal function change? There should be more clarification on why the RI is more predictive than GFR. Or, if it is a combination that is best, as stated in paragraph 8, that should be in the abstract conclusion.

Discussion, paragraph 11: “we had to adopt an animal model to study alveolar tissue samples.” Did you actually study an animal model? I did not see anything in the methods or results.

Reviewer #3: The authors described early changes in pulmonary function and intrarenal haemodynamics with patients with T2DM less than 12 years duration. Their findings are interesting and the study is well designed. Few things needed to be addressed:

- EDV abbreviation is not included in text of manuscript.

- PED and EDV had been used for same concept and need to be changed.

- Renal index defined as retobulbar hemodynamic which is confusing and not accurate; Also intraocular pressure measurement had been mentioned to be evaluated. This do not sound right in the context of the manuscript with no data provided to support.

-Table 1 indicate 33 healthy subjects and 37 patients in diabetic group. However, the text indicate 96 patients enrolled in the study. Authors should address the discrepancy and repeat the statistics on whole panel of patients group in table 1.

6. PLOS authors have the option to publish the peer review history of their article (what does this mean?). If published, this will include your full peer review and any attached files.

Reviewer #1: No

Reviewer #2: No

Reviewer #3: No

---

## [Author Response · Author response to Decision Letter 0]

30 Sep 2019

Reviewer 1

We spell-checked the manuscript, took out the redundancy, and also replaced few terminologies per Review #1’s comments. By doing so, we invited our colleague Ms. JJ JIAJIA Yu a more experienced in English language to help us with the correction.

Reviewer 2

Reviewer 2’s comments were taken seriously and we will try to put through a thorough revision. Meanwhile, we’d like to clarify Reviewer 2’s comments as following. 

1. Grammar

① Author’s page: “Corresponding autuo” should read Corresponding author

②Discussion, 4th paragraph: “that idea” should read the idea

③Discussion, 4th paragraph: “Study noted weak associations…” should read One study

④Discussion, 8th paragraph: “sufficient statistical power, we able…” should read we were able

2. Introduction

① During the early period, the renal function is normal, the GFR had increaded relative to healthy people, but the GFR is in the normal range.

3. Methods

① We had researched the retrobulbar haemodynamics 2 years, so we make a mistake.

4. Results

① The renal function parameters are positively correlated with HbA1c and diabetes duration. the renal function will get worse and worse as the time (diabetes duration) went on, The worse renal function is accompanied by increased renal function parameters. The renal function is different from renal function parameters, two different concepts.

② Pulmonary functional parameters are different from Pulmonary function, the worse Pulmonary function is accompanied by depressed pulmonary functional parameters. So the pulmonary function parameters and renal function parameters are negatively correlated rather than the pulmonary function and renal function are negatively correlated.

5. Discussion

① Our study recruited the T2DM patients with normal renal function, so we choosed the T2DM patients during the early period, because during the early period, very few patients would appeared the abnormal renal function, although, the renal function parameters were increased.

② During the early period, the renal function parameters was higher than healthy people, but the renal function was normal. GFR as a important renal function parameters will be increased, so we can choose it as an predictor of renal function change. But we can not state the value of combination of GFR and RI, because in our study we do not research the combination, they will be studied in future.

③ Our group found the value of RI, and choosed the patients to study, discover the phenomenon and study the mechanism in animals and cells, we have build the rat model to study, in dddition, more and more studies had found the lung will get abnormal as following.

Heyuan Wang, Wei Wu, Guixia Wang, et al. Protective effect of ginsenoside Rg3 on lung injury in diabetic rats. J Cell Biochem. 2018;1-8. DOI: 10.1002/jcb.27601

Fang Zhang1, Fei Yang, Hongmei Zhao, et al. Curcumin alleviates lung injury in diabetic rats by inhibiting NF-κB pathway. doi: 10.1111/1440-1681.12438

Reviewer 3

Reviewer 3’s comments were taken seriously and we will try to put through a thorough revision. Meanwhile, we’d like to clarify Reviewer 3’s comments as following. 

① We have corrected the PED, we mistaked the EDV. 

② Renal index included renal function parameters and intrarenal RI

③ We had researched the retrobulbar haemodynamics 2 years, so we make a mistake.

④ We have corrected the table1, 96 patients.

---

## [Editor Report · Decision Letter 1]

25 Oct 2019

Early Changes in Pulmonary Function and Intrarenal Haemodynamics and the Correlation between These Sets of Parameters in Patients with T2DM

PONE-D-19-16612R1

Dear Dr. jia,

We are pleased to inform you that your manuscript has been judged scientifically suitable for publication and will be formally accepted for publication once it complies with all outstanding technical requirements.

With kind regards,

Alon Harris

Academic Editor

PLOS ONE
---

## [Editor Report · Acceptance letter]

4 Dec 2019

PONE-D-19-16612R1 

Early Changes in Pulmonary Function and Intrarenal Haemodynamics and the Correlation between These Sets of Parameters in Patients with T2DM 

Dear Dr. Jia:

I am pleased to inform you that your manuscript has been deemed suitable for publication in PLOS ONE. Congratulations! Your manuscript is now with our production department. 

With kind regards,

on behalf of

Dr. Alon Harris 

Academic Editor

PLOS ONE